# Elective and nonelective cesarean section and obesity among young adult male offspring: A Swedish population–based cohort study

**Viktor H. Ahlqvist**[1], **Margareta Persson**[2], **Cecilia Magnusson**[1,3], **Daniel Berglind**[1,3]*

**1** Department of Global Public Health Sciences, Karolinska Institutet, Stockholm, Sweden, **2** Department of Nursing, Umeå University, Umeå, Sweden, **3** Centre for Epidemiology and Community Medicine, Region Stockholm, Stockholm, Sweden

* daniel.berglind@ki.se

**Data Availability Statement:** Swedish secrecy law prohibits us from making register data publicly available. The data supporting our findings were used under license and ethical approval for the

## Abstract

### Background

Previous studies have suggested that cesarean section (CS) is associated with offspring overweight and obesity. However, few studies have been able to differentiate between elective and nonelective CS, which may differ in their maternal risk profile and biological pathway. Therefore, we aimed to examine the association between differentiated forms of delivery with CS and risk of obesity in young adulthood.

### Methods and findings

Using Swedish population registers, a cohort of 97,291 males born between 1982 and 1987 were followed from birth until conscription (median 18 years of age) if they conscripted before 2006. At conscription, weight and height were measured and transformed to World Health Organization categories of body mass index (BMI). Maternal and infant data were obtained from the Medical Birth Register. Associations were evaluated using multinomial and linear regressions. Furthermore, a series of sensitivity analyses were conducted, including fixed-effects regressions to account for confounders shared between full brothers. The mothers of the conscripts were on average 28.5 (standard deviation 4.9) years old at delivery and had a prepregnancy BMI of 21.9 (standard deviation 3.0), and 41.5% of the conscripts had at least one parent with university-level education.

Out of the 97,291 conscripts we observed, 4.9% were obese (BMI $\geq$ 30) at conscription. The prevalence of obesity varied slightly between vaginal delivery, elective CS, and nonelective CS (4.9%, 5.5%, and 5.6%, respectively), whereas BMI seemed to be consistent across modes of delivery. We found no evidence of an association between nonelective or elective CS and young adulthood obesity (relative risk ratio 0.96, confidence interval 95% 0.83–1.10, $p$ = 0.532 and relative risk ratio 1.02, confidence interval 95% 0.88–1.18, $p$ = 0.826, respectively) as compared with vaginal delivery after accounting for prepregnancy maternal BMI, maternal diabetes at delivery, maternal hypertension at delivery, maternal smoking, parity, parental education, maternal age at delivery, gestational age, birth weight standardized according to gestational age, and preeclampsia. We found no evidence of an

current study. Readers interested in obtaining microdata or replicating our study may seek similar approvals and inquire through Statistics Sweden. For further advice see: https://www.scb.se/en/services/guidance-for-researchers-and-universities/.

**Funding:** This work was supported by the Stockholm County Council [ALF 20180266 to DB]. The funders had no role in study design, data collection and analysis, decision to publish, or preparation of the manuscript.

**Competing interests:** The authors have declared that no competing interests exist.

**Abbreviations:** BMI, body mass index; CI, confidence interval; CS, cesarean section; IQR, interquartile range; MBR, Medical Birth Register; OR, odds ratio; RRR, relative risk ratio; WHO, World Health Organization.

association between any form of CS and overweight (BMI $\geq$ 25) as compared with vaginal delivery. Sibling analysis and several sensitivity analyses did not alter our findings. The main limitations of our study were that not all conscripts had available measures of anthropometry and/or important confounders (42% retained) and that our cohort only included a male population.

## Conclusions

We found no evidence of an association between elective or nonelective CS and young adulthood obesity in young male conscripts when accounting for maternal and prenatal factors. This suggests that there is no clinically relevant association between CS and the development of obesity. Further large-scale studies are warranted to examine the association between differentiated forms of CS and obesity in young adult offspring.

## Trial registration

Registered as observational study at ClinicalTrials.gov Identifier: NCT03918044.

## Author summary

### Why was this study done?

- The global prevalence of cesarean deliveries is increasing, and clarification of any harmful consequences for offspring health is warranted.

- There is evidence of an association between delivery by cesarean section and obesity, but it may be driven by unmeasured confounding.

- It has been suggested that elective but not nonelective cesarean section is associated with offspring obesity, but the knowledge base to support this notion is scant.

### What did the researchers do and find?

- A register-based total-population cohort of 97,291 Swedish males was followed up for levels of overweight and obesity in early adulthood using objective measures of weight and height at military conscription. Sibling comparison was used to address bias from familial factors.

- Delivery by cesarean section, regardless of whether elective or nonelective, was not associated with overweight or obesity among Swedish men in young adulthood.

### What do these findings mean?

- Mode of delivery may not be an important factor in the origins of overweight and obesity.

- Cesarean section may not serve a role in the obesity epidemic and, as such, should not be a target for intervention when attempting to reduce the burden of obesity.

- Future research should include female offspring when examining the role of cesarean section in obesity to critically evaluate any sex-specific role of cesarean section in obesity.

## Introduction

Globally [1], and in Sweden [2], there has been an unprecedented increase in the prevalence of cesarean deliveries since the early 1990s. Between 1990 and 2014 the world prevalence of cesarean section (CS) increased by 285% (6.7% versus 19.1% of all births) [1], albeit with large regional disparities. The indications for CS are many, and CS is often warranted to avoid fetal and/or maternal morbidity and mortality [3]. However, indications for CS are to some degree subjective [4], and changes in maternal risk profiles do not explain the increased prevalence of CS [5,6]. Maternal preference and/or fear of childbirth has been described to be a contributing factor to the increase in CS rates [7–9]. Notably, the World Health Organization (WHO) states that, on a population level, rates of CS higher than 10%–15% are not associated with additional reductions in maternal, neonatal, and infant mortality rates [3].

The large increases of CS have sparked an interest in its long-term effects on offspring health [3]. Indeed, CS has been associated with various negative outcomes—e.g., asthma [10,11], allergic rhinitis [11], and food allergies [12], as well as overweight and obesity [10,13–16]. Proposed mechanisms explaining the observed association between CS and subsequent morbidity in the offspring include, but are not limited to, hormonal surge [17], lack of stress exposure [17], DNA methylation [18–20], and microflora transmission (hygiene hypothesis) [20,21].

The association between CS and offspring overweight and obesity has recently received further attention [10,15,22–25]. Notably, the unprecedented increase in CS has occurred at the same time as the obesity epidemic [26], which may suggest a connection between the two trends. However, as previous studies have noted [22,23,25], the association between CS and obesity may be mainly driven by unmeasured confounding. Yuan and colleagues [15] conducted a careful investigation aiming to control for such confounding, but they still observed an association between CS and offspring obesity. Yuan and colleagues [15] did not, however, differentiate between elective and nonelective CS. To the best of our knowledge, only a few studies have investigated effects of elective and nonelective CS separately [23–25]. This is unfortunate because the indications for and the obstetric risks of CS may depend on the type of CS, which could make the confounding structure differ importantly between the types of CS. Notably, one of the likely candidates for unmeasured confounding is confounding by indication, which could inflate the risk for obesity in the nonelective CS group [23], given that nonelective CS may be more likely to be conducted on the basis of fetal indication and obstetric complications [27,28]. Additionally, elective and nonelective CS may differ in fetal stress exposure [29] and microflora transmission [30], which could have implications for associations to subsequent obesity. Furthermore, given the rise in CS [1], there could be different implications for public health in the presence of different risks of development of obesity depending on the type of CS delivery. Despite few studies differentiating between elective and nonelective CS, at 12 months [24] but not at 3 and 5 years of age [23], there have been reported

elevated risks of obesity in elective CS. However, large-scale longitudinal studies are warranted, and there is a necessity to examine whether this indicated association persists into young adulthood.

## Objective

Here, we aim to examine the association between differentiated forms of delivery with CS and risk of obesity in young adulthood in a large, total-population sample of male conscripts based on objectively measured anthropometry. We evaluate the role of confounding by indication by separately studying elective and nonelective CS, carefully controlling for confounding factors including prepregnancy maternal body mass index (BMI), and by sibling comparisons, hence adjusting for shared familial factors.

## Methods

### Study design

A male total population–based cohort was constructed using the following Swedish population–based registries: (1) the Swedish Military Service Conscription Registry [31], (2) the Swedish Medical Birth Register (MBR) [32], (3) the Multi-Generation Register [33], and (4) the population and housing censuses from 1970 and 1990 [33]. Registers were linked via a personal identification number, a unique ID code assigned to each Swedish resident at birth. Furthermore, all full brothers and parents were identified and matched using a unique family identification number through the Multi-Generation Register. The study was approved by the Regional Ethical Review Board, Stockholm (Dnr: 2016/1445-31/1). The study protocol and statistical analysis plan were registered on April 17, 2019, at clinicaltrials.gov (NCT03918044) to increase transparency and reduce the risk for post hoc analysis.

### Study population

Using the MBR, which contains validated birth data on approximately 99% of the Swedish population [32], all male singletons born between 1982 and 1987 available in the MBR were sampled ($n$ = 300,344) (Fig 1). The first exclusion criterion was not having available information on the mode of delivery ($n$ = 16,970). The sampled singletons were then matched to their conscription data if they conscripted before 2006 using the Swedish Military Service Conscription Registry ($n$ = 229,632). During the study period, conscription was mandatory by law in Sweden for all male citizens up to the age of 47 [34]. The study period was selected because of the changes in conscription procedures occurring after the set period. During the study period, males could only be exempt from conscriptions (requiring state approval) if they suffered from chronic disease or severe handicap [35]. Except for in sensitivity analyses, we excluded those with extreme values of weight ($\leq$40 or $\geq$150 kg), height ($\leq$150 or $\geq$210 cm), and/or BMI ($\leq$15 or $\geq$60 kg/m$^2$) at conscription ($n$ = 105) in accordance with previous studies [36]. All analyses were conducted as complete-case analyses because we a priori hypothesized any missing data to be missing completely at random and potentially missing not at random in a few cases; therefore, we excluded all individuals with missing height and/or weight at conscription ($n$ = 76,410) and/or those with missing information on other characteristics of interest (e.g., maternal BMI and maternal smoking) ($n$ = 55,826) (S1 Table). In total, 97,291 male conscripts with a median age of 18 years at follow-up (i.e., conscription) remained eligible for analysis and were included in the final analytical sample. In addition, we identified 9,676 matchable full brothers.

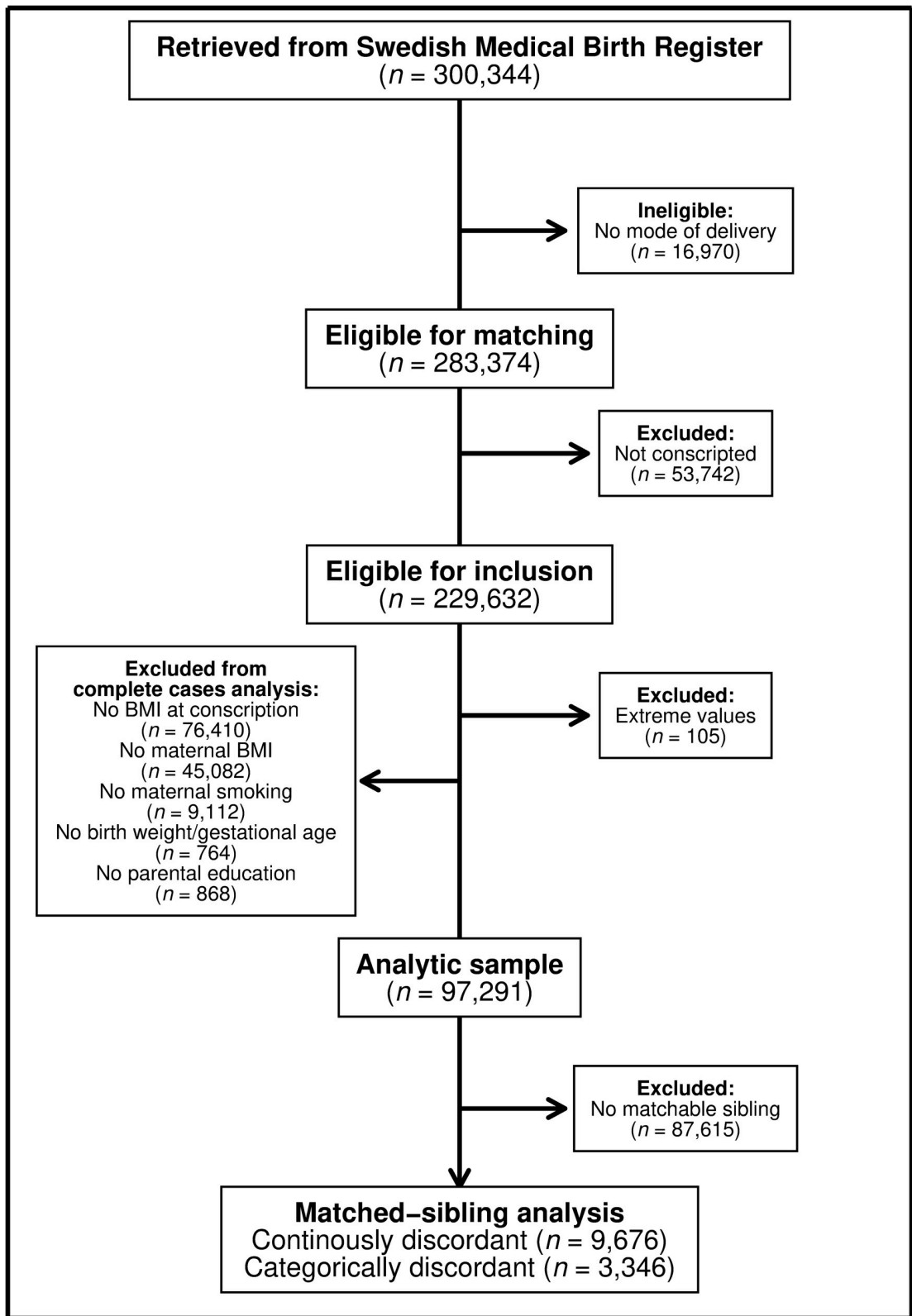

**Fig 1. Flowchart of the derivation of the analytical sample.** BMI, body mass index.

### Exposure—Mode of delivery

Using the Swedish MBR, we obtained information on recorded mode of delivery (vaginal or cesarean delivery), which was supplemented with information on indication for cesarean delivery (elective or nonelective), which yielded our primary trichotomized exposure coded as (1) vaginal delivery, (2) delivery by elective CS, and (3) delivery by nonelective CS. In accordance with the reporting standard in the MBR, we define elective CS as prelabor CS and nonelective CS as CS after the onset of labor. There was no alteration in the reporting procedure of elective or nonelective CS in the MBR during the period of study. Additionally, to facilitate comparison with previous studies, we report the dichotomized exposure: (1) vaginal delivery and (2) delivery by any form of CS. Finally, in a post hoc sensitivity analysis, the exposures (1) vaginal delivery, (2) instrumental vaginal delivery (forceps or vacuum extraction), (3) delivery by elective CS, and (4) delivery by nonelective CS were used to diminish the assumption of homogeneity in vaginal deliveries.

### Outcome—Young adulthood BMI

The primary outcome was obesity defined according to categories of BMI (kg/m$^2$) categorized using WHO's [37] standards: underweight (BMI < 18.5), normal weight (BMI 18.5–24.9), overweight (BMI 25–29.9), and obese (BMI ≥ 30). Although our aim is to examine obesity, we quantify associations between CS and all categories of BMI as compared with normal weight BMI. Weight in kilograms and height in centimetres were measured at conscription using a standardized scale and a stadiometer [36] under supervision of a nurse or physician [38]. To facilitate comparisons with previous studies, we also examined the odds of obesity (BMI ≥ 30) using all other BMI categories as reference outcome (BMI < 30). Finally, we estimated the association between our primary exposure (vaginal/elective CS/nonelective CS) and BMI at conscription as a continuous variable.

### Confounders

We considered a series of confounders that have been previously associated with cesarean delivery and metabolic or adiposity-related factors in offspring. From the MBR data on pre-pregnancy maternal BMI [39,40] (continuous), maternal diabetes at delivery [41,42] (yes/no), maternal hypertension at delivery [43] (yes/no), self-reported maternal smoking [44,45] at the commencement of pregnancy (nonsmoker, 1–9 cig/day, ≥10 cig/day), parity [46,47] (treated as categorical), birth weight in grams [48,49] standardized according to week of gestational age using the total population as reference (continuous), preeclampsia [50,51] (ICD-8: 63703–63710 and ICD-9 642E-642G) (yes/no), gestational age [49,52] (continuous), and maternal age at delivery [53,54] (continuous) were collected. Using the population and housing censuses, we identified the highest level of paternal and maternal education [55,56] (categorical) to serve as a proxy for household socioeconomic status.

### Statistical analyses

We descriptively present the distribution of the outcomes and confounders over the total analytic cohort and over the primary exposures (vaginal, elective CS, and nonelective CS) using appropriate measures of central tendency and dispersion. For our main analysis, we employed multinomial logistic regression to estimate crude and confounder-adjusted relative risk ratios (RRRs) with 95% confidence intervals (CIs). All standard errors were estimated using the robust (sandwich) method to account for the correlation between brothers [57]. All statistical analyses were performed using Stata 15.1 (Stata Corp, College Station, TX, United States).

## Sensitivity analyses

We descriptively present the available characteristics of the individuals not participating in conscription and those excluded due to missing data and compare these to the conscripted and analytic sample, respectively, using $\chi^2$ test (categorical), $t$ test (continuous), and Wilcoxon rank-sum test (continuous skewed). In the sensitivity analysis, we introduced several multinomial logistic models. First, a model in which we introduced a cubic transformation of maternal age and maternal BMI, in addition to the untransformed factor, to relax the assumption of linearity was used. Second, a model in which we adjusted for gestational weight gain [58] in a subset of individuals with available measures of maternal BMI at delivery ($n$ = 96,050) standardized by categories of BMI and gestational week according to Swedish reference values [59] was used. Third, using information on previous cesarean deliveries recorded after 1973, we adjusted for whether a mother had ever had a CS before (binary). Previous cesarean delivery may impact the association between a subsequent cesarean delivery and obesity [15]. Fourth, we included those with available data previously excluded for extreme values at conscription ($n$ = 62) to ascertain that our exclusion did not alter our primary findings. In our final multinomial model, we examined the association between any form of cesarean delivery and BMI categories. For our secondary outcomes, we employed logistic regression to examine the association between our primary exposure and obesity and linear regression, treating BMI as a continuous outcome. Furthermore, we employed multinomial logistic regression with fixed effects (conditional) [60] and fixed-effects linear regression [22] in a subset of 3,346 and 9,676 discordant full brothers, respectively, to account for unmeasured familial confounding (genetic and environmental) [61]. The fixed-effects regressions were adjusted for the same factors as our main analysis, excluding highest parental education, which did not vary between full brothers. In the post hoc analysis, we examine the influence of CS on ordered BMI categories, employing a generalized ordered logit model that was relaxed of proportionality assumptions [62], and we further relaxed linearity of all continuous factors (maternal prepregnancy BMI, maternal age, birth weight standardized according to gestational age, and gestational age) using restricted cubic splines with five knots at Harrell's recommended positions [63]: 5th, 27.5th, 50th, 72.5th, and 95th percentiles. Because previous validity reports [32] have noted that there may be some misclassification of the type of CS in the MBR and that this misclassification is primarily present in preterm deliveries that were misclassified as elective while being nonelective, we repeated our main analysis excluding those born preterm (<37 full weeks of gestation) and repeated our main analysis restricted to those born at term (≥37 and <42 weeks of gestation). Finally, as gestational age may act as a collider under certain causal pathways, we repeated our main analysis, excluding adjustment for gestational age.

## Compliance with ethical standards

The study was approved by the Regional Ethical Review Board, Stockholm (Dnr: 2016/1445-31/1). The requirement to obtain informed consent was waived by the Regional Ethical Review Board, Stockholm (Dnr: 2016/1445-31/1). All research was performed in accordance with relevant guidelines/regulations.

## Results

### Descriptive statistics

In our cohort of 97,291 conscripts, we observed that 4.9% were obese at conscription (Table 1). The prevalence of obesity among those born by vaginal delivery was 4.9% (CI 95% 4.7–5.0), which was not statistically significantly different from those born by elective CS

**Table 1. Sample characteristics of the full analytical cohort by mode of delivery.**

| Sample characteristics | Total (N = 97,291) | Vaginal delivery (N = 89,024) | Elective cesarean section (N = 4,147) | Nonelective cesarean section (N = 4,120) |
|---|---|---|---|---|
| **Offspring characteristics** | | | | |
| Age at conscription (years), median (IQR) | 18 (18, 18) | 18.0 (18, 18) | 18 (18, 18) | 18 (18, 18) |
| BMI category at conscription, No. (%) | | | | |
| Underweight | 5,945 (6.1) | 5,491 (6.2) | 224 (5.4) | 230 (5.6) |
| Normal weight | 71,511 (73.5) | 65,479 (73.6) | 3,044 (73.4) | 2,988 (72.5) |
| Overweight | 15,041 (15.5) | 13,720 (15.4) | 650 (15.7) | 671 (16.3) |
| Obese | 4,794 (4.9) | 4,334 (4.9) | 229 (5.5) | 231 (5.6) |
| BMI (kg/m$^2$) at conscription, mean (SD) | 22.8 (3.7) | 22.8 (3.6) | 23.0 (3.8) | 23.1 (3.8) |
| Birth weight (g), mean (SD) | 3,615.9 (518.3) | 3,633.1 (499.4) | 3,444.0 (554.7) | 3,416.8 (754.1) |
| Weeks of gestation, mean (SD) | 39.5 (1.6) | 39.6 (1.5) | 38.2 (1.3) | 39.0 (2.6) |
| **Maternal characteristics** | | | | |
| Maternal age at birth (years), mean (SD) | 28.5 (4.9) | 28.4 (4.9) | 30.6 (5.3) | 28.7 (5.2) |
| Maternal prepregnancy BMI, mean (SD) | 21.9 (3.0) | 21.9 (3.0) | 22.3 (3.4) | 22.4 (3.2) |
| Maternal BMI category, No. (%) | | | | |
| Underweight | 7,450 (7.7) | 6,847 (7.7) | 313 (7.5) | 290 (7.0) |
| Normal weight | 77,080 (79.2) | 70,875 (79.6) | 3,139 (75.7) | 3,066 (74.4) |
| Overweight | 10,933 (11.2) | 9,727 (10.9) | 562 (13.6) | 644 (15.6) |
| Obese | 1,828 (1.9) | 1,575 (1.8) | 133 (3.2) | 120 (2.9) |
| Parity, median (IQR) | 2.0 (1.0, 2.0) | 2.0 (1.0, 2.0) | 2.0 (1.0, 3.0) | 1.0 (1.0, 2.0) |
| Maternal diabetes mellitus, No. (%) | 463 (0.5) | 354 (0.4) | 77 (1.9) | 32 (0.8) |
| Maternal hypertension, No. (%) | 204 (0.2) | 164 (0.2) | 16 (0.4) | 24 (0.6) |
| Preeclampsia, No. (%) | 1,515 (1.6) | 1,162 (1.3) | 103 (2.5) | 250 (6.1) |
| Maternal smoking at the commencement of pregnancy, No. (%) | | | | |
| Not smoking | 70,814 (72.8) | 64,915 (72.9) | 3,068 (74.0) | 2,831 (68.7) |
| 1–9 cig/day | 16,677 (17.1) | 15,163 (17.0) | 683 (16.5) | 831 (20.2) |
| ≥10 cig/day | 9,800 (10.1) | 8,946 (10.0) | 396 (9.5) | 458 (11.1) |
| **Socioeconomic factor** | | | | |
| Highest parental education, No. (%) | | | | |
| Primary education | 8,416 (8.7) | 7,635 (8.6) | 398 (9.6) | 383 (9.3) |
| Secondary education | 48,509 (49.9) | 44,538 (50.0) | 1,938 (46.7) | 2,033 (49.3) |
| University education | 40,366 (41.5) | 36,851 (41.4) | 1,811 (43.7) | 1,704 (41.4) |

Abbreviations: BMI, body mass index; cig, cigarette; IQR, interquartile range; No., number; SD, standard deviation

(5.5%, CI 95% 4.8–6.2, $p$ = 0.057). The prevalence of obesity among those born by nonelective CS was 5.6% (CI 95% 4.9–6.3), which was statistically significantly higher as compared with those born by vaginal delivery ($p$ = 0.032). The mean BMI was lower among those born by vaginal delivery (22.8, CI 95% 22.8–22.8) as compared with those born by elective CS (23.0, CI 95% 22.9–23.1, $p$ = 0.010) or nonelective CS (23.1, CI 95% 22.9–23.2, $p$ < 0.001). Although statistically different (all $p$ < 0.05), most covariates did not vary substantially with mode of delivery. However, those with vaginal delivery had higher birth weight, lower maternal BMI, and lower occurrence of maternal prepregnancy obesity. The elective CS group had higher occurrence of university-level parental education and of maternal diabetes mellitus. The nonelective CS group had a higher occurrence of preeclampsia and maternal smoking at the commencement of pregnancy.

**Table 2. Cesarean section deliveries and their associations to obesity, overweight, and underweight in offspring.**

| | Crude | | | | | | | | |
|---|---|---|---|---|---|---|---|---|---|
| | Underweight versus normal weight | | | Overweight versus normal weight | | | Obese versus normal weight | | |
| **Mode of delivery** | **RRR** | **95% CI** | **p** | **RRR** | **95% CI** | **p** | **RRR** | **95% CI** | **p** |
| Vaginal | 1 | - | - | 1 | - | - | 1 | - | - |
| Elective cesarean section | 0.88 | 0.76–1.01 | 0.064 | 1.02 | 0.93–1.11 | 0.669 | 1.14 | 0.99–1.30 | 0.069 |
| Nonelective cesarean section | 0.92 | 0.80–1.05 | 0.220 | 1.07 | 0.98–1.17 | 0.113 | 1.17 | 1.02–1.34 | 0.027 |
| | Adjusted[a] | | | | | | | | |
| | Underweight versus normal weight | | | Overweight versus normal weight | | | Obese versus normal weight | | |
| | **RRR** | **95% CI** | **p** | **RRR** | **95% CI** | **p** | **RRR** | **95% CI** | **p** |
| Vaginal | 1 | - | - | 1 | - | - | 1 | - | - |
| Elective cesarean section | 0.88 | 0.76–1.01 | 0.079 | 0.99 | 0.90–1.08 | 0.818 | 1.02 | 0.88–1.18 | 0.826 |
| Nonelective cesarean section | 0.94 | 0.81–1.08 | 0.359 | 0.99 | 0.90–1.08 | 0.764 | 0.96 | 0.83–1.10 | 0.532 |

Empty cells (-) indicate reference group.

[a]Adjusted for prepregnancy maternal BMI, maternal diabetes at delivery, maternal hypertension at delivery, maternal smoking, parity, parental education, maternal age at delivery, birth weight standardized according to gestational age, preeclampsia, and gestational age.

Abbreviations: BMI, body mass index; CI, confidence interval; RRR, relative risk ratio

## Mode of delivery and offspring obesity

In our primary analysis (Table 2), there was no statistically significant association between elective CS and obesity (RRR 1.14, CI 95% 0.99–1.30, $p$ = 0.069), whereas a similar association between nonelective CS and obesity differed from unity (RRR 1.17, CI 95% 1.02–1.34, $p$ = 0.027) as compared with vaginal delivery. We observed no association between elective or nonelective CS and obesity (RRR 1.02, CI 95% 0.88–1.18, $p$ = 0.826 and RRR 0.96, CI 95% 0.83–1.10, $p$ = 0.532, respectively) as compared with vaginal delivery when accounting for pre-pregnancy maternal BMI, maternal diabetes at delivery, maternal hypertension at delivery, maternal smoking, parity, parental education, maternal age at delivery, gestational age, birth weight standardized according to gestational age, and preeclampsia. Neither elective nor non-elective CS were associated with overweight at conscription as compared with vaginal delivery.

## Sensitivity analysis

Characteristics in individuals who were and were not conscripted did not differ materially overall, although differences were statistically significant in our large sample (all $p < 0.05$, S1 Table). Similarly, although every characteristic except mean BMI and maternal preeclampsia were statistically different ($p < 0.05$), there was no major discrepancy between those with incomplete records and those retained in the analytical cohort. However, we noted that parental university education was more common among the conscripted as compared with the not conscripted (38.4% versus 29.7%, $p < 0.05$) and more common among the retained analytical cohort as compared with those with incomplete records (41.5% versus 36.1%, $p < 0.05$).

The stratification of vaginal deliveries into vaginal instrumental deliveries and vaginal deliveries did not alter our main findings; there was no meaningful change in any estimate as compared with our main analysis (S2 Table).

Our further sensitivity analyses (S3 Table), in which we (1) relax linearity assumptions of confounders, (2) account for standardized gestational weight gain, (3) account for a history of cesarean delivery, and (4) include those previously excluded at conscription, did not differ to any meaningful extent from our main analysis. In our last multinomial model (S4 Table),

when treating all CSs as one group, we did not observe any association between CS and obesity after accounting for confounders (RRR 0.98, CI 95% 0.89–1.09, *p* = 0.775) as compared with vaginal delivery. When treating obesity as a binary outcome (S5 Table), there was no association between elective or nonelective CS and obesity in our fully adjusted model (RRR 1.02, CI 95% 0.88–1.18, *p* = 0.772 and RRR 0.96, CI 95% 0.84–1.11, *p* = 0.603, respectively) as compared with vaginal delivery. There was no association between elective (mean difference: 0.05, CI 95% −0.06 to 0.16, *p* = 0.338) or nonelective CS (mean difference: 0.02, CI 95% −0.09 to 0.14, *p* = 0.675) and continuous BMI when accounting for proposed confounders as compared with vaginal delivery (S6 Table). When employing multinomial fixed-effects regression, in a subset of discordant siblings, we observed a statistically significant association (*p* = 0.04) between nonelective CS and overweight (RRR 1.99, CI 95% 1.05–3.77) as compared with vaginal delivery (S7 Table). No other fixed-effects regression (linear or multinomial logistic) differed meaningfully from our main analysis (S7 Table and S8 Table).

There was no difference between our main analyses and when excluding those born preterm, restricting to those born at term, not adjusting gestational age (S9 Table), or treating BMI categories as an ordered outcome (S10 Table).

## Discussion

### Main findings

In this longitudinal cohort of 97,291 male conscripts, we found that there was no association between elective or nonelective CS and young adulthood obesity when accounting for possible confounding factors and conducting several predefined sensitivity analyses as compared with vaginal delivery. To the best of our knowledge, we have presented the largest and most comprehensive differentiation between elective and nonelective CS so far.

### Comparison with previous research

In contrast with previous research [10,13,15,16], we did not observe any association between CS and young adulthood obesity. This could be explained by previous studies' limited ability to adjust for maternal prepregnancy BMI [16], small sample sizes [10,13,15,16], and/or inability to differentiate between elective/prelabor and nonelective/acute CS [10,16]. To the best of our knowledge, the largest previous study on CS and offspring overweight/obesity that differentiated between types of CS was conducted on children (5 years old), with only 145 exposed cases, and described a significant association between CS on maternal request and overweight/obesity (odds ratio [OR] 1.18, CI 95% 1.00–1.41) [64]. In contrast, a meta-analysis [13] suggested that there was no association between prelabor CS and obesity in adulthood. However, this analysis was limited to very few participants (prelabor CS = 252 and vaginal delivery = 17,506).

Although a majority of previous studies have had limitations, studies with less uncertainty have suggested that there may be an attenuation with age in the association between CS and obesity [15,23]. The attenuation by age could potentially explain our findings. However, childhood obesity not persisting into young adulthood or adulthood may be of less relevance for clinical manifestation of obesity-related consequences [65–67].

Contrary to our findings, a recent study [24] on very young children (12 months old) suggested a strong association between elective CS and overweight (OR 2.01, CI 95% 1.13–3.58) but no such association for emergency CS (OR 1.08, CI 95% 0.66–1.76). The authors hypothesize that this may be a function of lack of fetal stress exposure induced by the onset of labor, which may be absent in elective CS, and that emergency CS may be exposed to maternal microbiota to a greater extent than elective CS. However, adjusting for intrapartum antibiotics

did not alter their findings, suggesting that microbiota exposure did not explain the low risk in the emergency CS [24]. Contrary to the suggestion that CS supports the hygiene hypothesis, there have been suggestions that CS is a risk factor for childhood obesity that is independent of second- and third-trimester antibiotic use [68], although antibiotic use may also independently influence childhood obesity [68].

In addition to the aforementioned discrepancies between our study and previous studies, it should be noted that our cohort was born between 1982 and 1987, which could have implications for the comparability to more recent birth cohorts [24]. However, several studies conducted on younger children with more recent birth years have suggested null associations similar to those we observed in our cohort [22,23,25].

In our cohort, only accounting for maternal prepregnancy BMI attenuated the association between elective or nonelective CS and obesity by 15% and 16%, respectively (RRR 1.14 versus 0.99 and 1.17 versus 1.01) (S11 Table). Indeed, this was the strongest confounder in the association between either form of CS and obesity in our study. As obesity has a high heritability [69] and/or could be a function of fetal programming [70], it is plausible that taking maternal prepregnancy BMI into account captures both genetic predispositions transferred from mother to offspring and the fetal exposure to an obesogenic state in the mother. Using our fixed-effects linear regression, which accounts for familial confounding shared between brothers (e.g., obesogenic familial environment and obesity-related genetic traits), we did not observe any association between any form of CS and offspring obesity. Notably, we observed an association between nonelective CS and overweight in our multinomial discordant sibling sample. However, this may be explained by the small number of exposed cases and the possibility of residual confounding. Hence, we suspect there to be an association between nonelective CS and overweight as a function of confounding by medical indication for CS [23]. Indeed, if there is a causal effect of CS on overweight/obesity, an association should be present in elective CS as well [23].

Despite unnecessary CS being subject to some criticism [71], it is of interest to further study unnecessary CS and its possible association to offspring morbidity [72]. Although Swedish registers enable us to differentiate between elective and nonelective CS, they do not enable us to perform detailed classifications of CS using such classifications as the Robson classification [73], which may provide further insight into the potential role of CS in offspring health.

## Strengths and limitations

The strengths of our study lie in our longitudinal design, large sample size, ability to control for a series of possible confounders, and ability to differentiate between elective and nonelective CS. Furthermore, the usage of MBR and the Swedish Military Service Conscription Registry enabled us to retrieve objective and standardized measures of exposure (birth mode) and outcome (BMI), limiting measurement error and information bias.

Despite the strengths of our study, there are several limitations that should be acknowledged. First, although conscription was mandatory by law under the period of study, there may be selection bias present in our cohort because those offspring with severe health conditions are not eligible for conscription [35]. Despite the potential for selection bias, we did not observe major differences, albeit most factors are statistically significantly different ($p < 0.05$), between our analytic sample and the excluded populations, with the exception of highest parental education (S1 Table). However, caution is warranted when generalizing and interpreting our findings. Second, there are still potential unmeasured confounders that we failed to account for. Third, we were limited to any record of previous CS in the MBR after 1973, which may fail to capture some previous CSs and fail to fully account for the motives to

perform an elective CS. Fourth, the usage of registers limited us to a formal diagnosis of maternal conditions, which may only capture the most severe cases of maternal health. Furthermore, there may be some misclassification of nonelective CS among preterm deliveries [32]. However, our analysis excluding preterm deliveries and restricting to at-term deliveries was consistent with our main analysis, suggesting that this misclassification of nonelective CS does not influence our conclusion. Despite this, a certain proportion of misclassification is expected and could inflate the risk estimate of an offspring developing obesity in the elective CS group, if an offspring of a nonelective CS is at higher risk of developing obesity as compared with those with a true elective CS, given medical indications. Fifth, despite the ability to control for unmeasured confounding, sibling analysis has several limitations, such as increased bias from nonshared confounders and increased attenuation from measurement error, as compared with conventional analysis [57]. However, our collection of CS by medical records and standardized measurement of BMI reduces the impact of measurement error. Furthermore, by accounting for observed nonshared confounders, we limit any additional confounding to nonshared unobserved confounders. Despite this, caution is warranted when directly interpreting the estimates from sibling analysis. Finally, our cohort consisted of a male population, which limits the generalizability to females. However, studies including female populations have suggested no difference between males and females in the association between CS and obesity [14,15,23,25,64]. The generalizability of our study may be limited to countries with rates of CS similar to those in Sweden during the study period.

## Conclusions

We observed no association between elective or nonelective CS and young adulthood obesity in young male conscripts when accounting for maternal and prenatal factors. Furthermore, we note that most of the crude association between CS and obesity could be explained by maternal prepregnancy BMI. This suggests that there is no clinically relevant association between CS and the development of obesity. Further large-scale studies are warranted to examine the association between differentiated forms of CS and obesity in young adult offspring.

## Supporting information

**S1 Table. Descriptive characteristics of the total population, eligible population, and the step-by-step excluded population.**
(DOCX)

**S2 Table. Relative risk ratios associated with instrumental vaginal delivery, elective cesarean section, and nonelective cesarean section as compared with vaginal delivery of underweight, overweight, and obesity relative to normal weight.**
(DOCX)

**S3 Table. Sensitivity analysis on associations between elective and nonelective cesarean section as compared with vaginal delivery of underweight, overweight, and obesity relative to normal weight.**
(DOCX)

**S4 Table. Association between pooled cesarean section and underweight, overweight, and obesity as compared with normal weight.**
(DOCX)

**S5 Table. Association between mode of delivery and obesity status (BMI ≥ 30).** BMI, body mass index.
(DOCX)

**S6 Table. Linear association between mode of delivery and continuous body mass index.**
(DOCX)

**S7 Table. Relative risk ratios associated with elective cesarean section and nonelective cesarean section as compared with vaginal delivery of underweight, overweight, and obesity relative to normal weight in a subset of discordant full brothers.**
(DOCX)

**S8 Table. Fixed-effects linear association between mode of delivery and continuous body mass index in a subset of full brothers.**
(DOCX)

**S9 Table. Association between mode of delivery and underweight, overweight, and obesity as compared with normal weight, excluding those born before 37 weeks of gestation (preterm), restricting to those born between 37 weeks and 41 weeks and 6 days of gestation (at term), and not adjusting for gestational age.**
(DOCX)

**S10 Table. Crude and adjusted association between mode of delivery and underweight, overweight, and obesity as compared with normal weight using generalized ordered logit estimation.**
(DOCX)

**S11 Table. Association between mode of delivery and underweight, overweight, and obesity as compared with normal weight, only adjusting for maternal prepregnancy body mass index.**
(DOCX)

## Author Contributions

**Conceptualization:** Viktor H. Ahlqvist, Cecilia Magnusson, Daniel Berglind.

**Formal analysis:** Viktor H. Ahlqvist.

**Methodology:** Viktor H. Ahlqvist, Margareta Persson.

**Supervision:** Cecilia Magnusson, Daniel Berglind.

**Writing – original draft:** Viktor H. Ahlqvist.

**Writing – review & editing:** Margareta Persson, Cecilia Magnusson, Daniel Berglind.

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
