## [Decision Letter · Decision Letter 0]

30 Sep 2019

Dear Dr. Berglind,

Thank you very much for submitting your manuscript "Elective and non-elective cesarean section and the risk for obesity among young adult males: a population-based cohort study" (PMEDICINE-D-19-03196) for consideration at PLOS Medicine. 

Your paper was evaluated by a senior editor and discussed among all the editors here. It was also sent to three independent reviewers, including a statistical reviewer. The reviews are appended at the bottom of this email and the accompanying reviewer attachment from Reviewer 1 can be seen via the link below:

[LINK]

In light of these reviews, I am afraid that we will not be able to accept the manuscript for publication in the journal in its current form, but we would like to consider a revised version that addresses the reviewers' and editors' comments. Obviously we cannot make any decision about publication until we have seen the revised manuscript and your response, and we plan to seek re-review by one or more of the reviewers. 

We expect to receive your revised manuscript by Oct 14 2019 11:59PM. Please email us (plosmedicine@plos.org) if you have any questions or concerns.

We look forward to receiving your revised manuscript. 

Sincerely,

Louise Gaynor, MBBS PhD

Associate Editor 

PLOS Medicine

plosmedicine.org

1. PLOS Medicine requires that the de-identified data underlying the specific results in a published article be made available, without restrictions on access, in a public repository or as Supporting Information at the time of article publication, provided it is legal and ethical to do so. Please see the policy at 

http://journals.plos.org/plosmedicine/s/data-availability

and FAQs at 

http://journals.plos.org/plosmedicine/s/data-availability#loc-faqs-for-data-policy

Specifically, please provide a URL or email contact information from where data can be requested. Please note that the contact for data access cannot be one of the authors.

2. Title: Please include that the study was based on a Swedish population. Also, we suggest that you please change “risk for” to “association with” in the title.

3. Abstract: Please include the years during which the study took place, and length of follow up. Specifically, please include the relevant information on cohort birth years, and year of conscription.

4. Abstract: Please quantify the main results (with 95% CIs and p values). Specifically, please include p values for tests of association between C-section birth and obesity.

5. Abstract: Please include the specific dependent variables that are adjusted for in the adjusted analyses described.

6. Abstract: Please address the study implications without overreaching what can be concluded from the data; your study is observational and therefore causality cannot be inferred from your results. Please remove language that implies causality, such as “This suggests that there is no clinically relevant causal role of CS in the development of obesity and that an association between CS and offspring obesity may be a function of confounding.”

8. Methods: Please clarify the age(s) of the individuals at the time of conscription (when BMI was assessed).

9. Results: Please provide 95% CIs and p values for statistical tests in support of the assertions that the prevalence of obesity varied slightly between delivery modalities (lines 176-178) and that most covariates did not vary substantially with mode of delivery (lines 178-179).

10. Results: Line 186-187: Please revise the statement: “In our primary analysis, there was an insignificant association between elective CS and obesity (RRR 1.14, CI 95%: 0.99-1.30)...” to more accurately state that there was no significant association observed between elective CS and obesity.

11. Results: Line 188: Please describe which factors are adjusted for in the analysis of the association between CS and obesity in the adjusted models.

12. Results: Please provide p values for the sensitivity analyses, and tests for associations between mode of delivery and offspring obesity status.

13. Discussion: Lines 218-220, and at 293-295: Please revise the language to avoid any misleading assertion that the lack of any association identified in the present study serves as supporting evidence for the idea that associations identified in other studies are attributable to confounding of elective vs. non-elective c-sections. 

14. Discussion: Lines 251-252: Please clarify where the analyses of association between CS and obesity adjusted only for pre-pregnancy BMI are presented.

15. Discussion: Lines 274-276: Were any statistical tests done to support the assertion that characteristics did or did not differ between the analyzed and excluded populations? If so, please present these in the results, and sTable1.

16. Tables: sTable7: There is a footnote for “a” but no “a” in the table. Please update footnote “b” to reflect the adjusted factors in the model.

17. Thank you for including the STROBE checklist. Please revise the checklist to use section and paragraph numbers, rather than page numbers and line numbers. 

Comments from the reviewers:

Reviewer #1: See attachment

Michael Dewey

Reviewer #2: Ahlqvist et al. present a retrospective cohort study on the association between birth by Caesarean section and obesity in young adulthood among 97,000 Swedish males. I have detailed my comments below.

Major comments: 

I do believe that replication in science is good and necessary, and research does not have to be novel to be publication-worthy. However, for the association between CS and offspring obesity, there have been around 100 studies (28 studies in Kuhle et al. 2015, 35 studies in Darmasseelane et al. 2014, and about 40 primary studies published since) published to date that looked at the issue from a lot of different angles in a multitude of settings using different kinds of analyses; the majority of studies adjusted for the key confounder maternal pre-pregnancy weight and still found an association. Based on the available evidence, I think it is safe to say that there is a true association. The paper by Ahlqvist et al, while solid, does not offer anything groundbreakingly new that would make me question the presence of that association between CS and obesity. 

I have reviewed a number of manuscripts on the subject, and most papers, like the one under review, justify their existence with the claim that confounding has not been sufficiently considered in previous studies (not true), only to go on to either use the same set of confounders as most previous studies, or add one confounder to their models and remove another. Ahlqvist et al perform a sibling-analysis to account for unmeasured familial confounding (as has been done before for this association), but this type of analysis adds confounding by non-shared confounders, as the paper by Frisell et al. 2012 (reference 57 in the manuscript) points out. This bias is not acknowledged or discussed by the authors. 

The other claim to (near-)novelty that the authors make is that "only a few studies [3 studies are referenced] have investigated effects of elective and non-elective CS in separate". I cannot verify or refute that claim, but in the presence of approximately 100 studies on the association, the authors need to demonstrate how they arrived at this number - did they do a systematic review of the literature? The authors also should describe clearly what the definition of elective and non-elective CS is, and if that definition is used consistently in the database. 

Another issue is the external validity of the study, since the men in the study were born more than 30 years ago (1982 to 1987). As expected, the CS rate in the sample is very low at 8.5% and the obesity prevalence sits at 4.9%. Both prevalences are considerably lower than what we see today. With the CS rate between 20 and 30% in most Western countries, I would question whether the population of women who had an elective CS in the 1980s is in any way comparable to women undergoing elective CS these days. The authors try to claim external validity by stating "In addition to aforementioned discrepancies between our study and previous studies, it should be noted that our cohort was born between 1982 and 1987, which could have implications for the comparability to more recent birth cohorts (24). However, several studies conducted on younger children with more recent birth years have suggested similar null associations as those we observed in our cohort (22, 23, 25)." I disagree with that logic: Because more recent studies also had null findings, the current study has external validity?

Minor comments:

- Use abbreviation CS consistently throughout

- Line 60: "in the offspring" not "in offspring's"

- Line 72: "which could inflate the risk for obesity in the non-elective cesarean section" should read "which could inflate the risk for obesity in the non-elective CS group" or "which could inflate the risk for obesity in children born by non-elective CS"

- Lines 189, 204, 231, 252: Replace "&" with "and"

- I would probably forego the adjustment for gestational age as it may be a collider. See the paper by Wilcox et al. 2011 (https://www.ncbi.nlm.nih.gov/pubmed/21946386)

- Line 158-159: "First, a model where we introduced a cubic transformation of maternal age and maternal BMI, to relax the assumption of linearity." Transforming age and BMI doesn't "relax" the assumption, the assumption is always there and should be met for the analysis. I would suggest examining linearity with LOESS or GAM first and then transform (or not) in the main analysis and skipping that part of the sensitivity analysis.

- Line 186: "In our primary analysis, there was an insignificant association between elective CS and obesity [...]" There is no "insignificant" association. Should read: "The association was not statistically significant."

- Lines 223-226: "In contrast to previous research (10, 13, 15, 16), we did not observe any association between CS and young adulthood obesity. This could be explained by previous studies limited ability to adjust for maternal pre-pregnancy BMI (16), small sample sizes (10, 13, 15, 16), and/or inability to differentiate between elective/pre-labor and non-elective/acute CS (10, 16)." The authors should avoid handpicking a few of the many many studies to give the impression that the existing body of research has serious shortcomings.

Reviewer #3: Ali Khashan (University College Cork)

The present study was performed to examine the association between mode of delivery and the risk of offspring overweight and obesity in early adulthood. The study used data from the Swedish national registers and included males who with data on height and weight measured at conscription. The study results showed no evidence of an association between elective or emergency CS and the risk of overweight or obesity in male offspring with the relative risk ratios very close to one. This is an important study considering the limited evidence on the association between CS and the risk of obesity in young adults. The manuscript is very well written and the fact that the statistical analysis plan was registered before the analysis was performed is a strength in this paper. The lack of an association between CS and risk of overweight or obesity in early adulthood is an important finding for mothers and clinicians. A key limitation is the fact BMI was measured only once while one would have liked more than one measurement over time. The authors may wish to consider the following comments:

1) I am not sure if I missed this, but what was the age of children at the time of weight and height measurement? Considering that conscription was up to age 47, does this mean the outcome was measured at different age points for different persons? If this were the case, then age at the time the outcome was measured should be taken into account in the analysis. 

2) Although the sibling analysis is an important analysis in this type of study, it is usually used to assess potential familial confounding when there is an association. In other words, it is used to determine whether an observed association is potentially causal or due to shared familial confounding. In this study there was no evidence of an association between CS and offspring obesity, therefore the sibling analysis has no added value and in fact it could be misleading as readers may misinterpret the significant result in the sibling analysis. I would remove the analysis from the manuscript and clarify that because there was no evidence of an association in the cohort analyses, the sibling analysis was not performed. 

3) Although I understand the authors' approach of excluding persons with missing outcome data, excluding persons with missing maternal BMI and missing maternal smoking is not justified. I would include those persons in the analysis and use a missing data indicator for those two variables. Including those persons in the crude analysis would help the authors assess whether their assumption on missing data is accurate or not. 

4) The authors highlight the potential misclassification of CS in preterm births, have they considered performing an analysis excluding preterm birth to check whether this potential misclassification is an issue? This is an analysis worth performing.

5) This study adds to a body of evidence from Sweden, that we and others did, showing limited evidence, if any, of an association between CS and asthma, type 1 diabetes, autism, ADHD and psychosis and the authors cited our two recent papers on CS and child obesity (reference 23 and 25) and we have two papers in press showing almost similar results from New Zealand and UK cohorts. The authors may want to have a more detailed discussion of these negative findings in terms of clinical practice and the fact CS does not seem to cause child morbidity, when good quality data and robust statistical analyses are used. Another discussion point is the fact Sweden has a low CS rate compared to other high income countries. Is it possible that elective CS in Sweden is elective due to medical indications and that what causes child morbidity, including obesity, is the unnecessary elective CS? I doubt this is the case, however, it is worth discussing the generalisability of results from a country with low CS rate to countries with high CS rate.

[LINK]

---

## [Decision Letter · Decision Letter 1]

29 Oct 2019

Dear Dr. Berglind,

Thank you very much for re-submitting your manuscript "Elective and non-elective cesarean section and the association with obesity among young adult males: a Swedish population-based cohort study" (PMEDICINE-D-19-03196R1) for review by PLOS Medicine.

I have discussed the paper with my colleagues and the academic editor and it was also seen again by reviewers. I am pleased to say that provided the remaining editorial and production issues are dealt with we are planning to accept the paper for publication in the journal.

[LINK]

We look forward to receiving the revised manuscript by Nov 05 2019 11:59PM. 

Sincerely,

Clare Stone, PhD

Managing Editor 

PLOS Medicine

plosmedicine.org

Requests from Editors:

- I'd suggest removing "the association with" from the title, as there is no association ("... C section and obesity among young adult male offspring ..."). Apologies for making requested changes to the title again. 

- Please add some summary demographic details for the conscripts in the abstract?

- I think the Author Summary could benefit from some "what do our findings mean" points and could mention the need to study young female offspring, for example?

 -Please use square brackets in the main text for references

Comments from Reviewers:

Reviewer #1: The authors have addressed my points.

Michael Dewey

Reviewer #3: I would like to thank the authors for addressing my comments and making the relevant changes in the text. I have no further comments.

[LINK]

---

## [Editor Report · Decision Letter 2]

5 Nov 2019

Dear Dr. Berglind, 

On behalf of my colleagues and the academic editor, Dr. Ali S Khashan, I am delighted to inform you that your manuscript entitled "Elective and non-elective cesarean section and obesity among young adult male offspring: a Swedish population-based cohort study" (PMEDICINE-D-19-03196R2) has been accepted for publication in PLOS Medicine. 

PRODUCTION PROCESS

PRESS

PROFILE INFORMATION

Thank you again for submitting the manuscript to PLOS Medicine. We look forward to publishing it. 

Best wishes, 

Clare Stone, PhD

Managing Editor 

PLOS Medicine

plosmedicine.org